# Pancreatic Islet Cells Response to IFNγ Relies on Their Spatial Location within an Islet

**DOI:** 10.3390/cells12010113

**Published:** 2022-12-28

**Authors:** Marine De Burghgrave, Chloé Lourenço, Claire Berthault, Virginie Aiello, Adrian Villalba, Alexis Fouque, Marc Diedisheim, Sylvaine You, Masaya Oshima, Raphaël Scharfmann

**Affiliations:** 1Université Paris Cité, Institut Cochin, INSERM U1016, CNRS UMR 8104, 75014 Paris, France; 2GlandOmics, 41700 Cheverny, France; 3Diabetology Unit, Cochin Hospital, AP-HP, 75014 Paris, France

**Keywords:** Langerhans islets, pancreatic beta cells, inflammation, IFNγ, type 1 diabetes, beta cells heterogeneity, endocrine population

## Abstract

Type 1 diabetes (T1D) is an auto-immune disease characterized by the progressive destruction of insulin-producing pancreatic beta cells. While beta cells are the target of the immune attack, the other islet endocrine cells, namely the alpha and delta cells, can also be affected by the inflammatory milieu. Here, using a flow cytometry-based strategy, we compared the impact of IFNγ, one of the main cytokines involved in T1D, on the three endocrine cell subsets isolated from C57BL/6 mouse islets. RNA-seq analyses revealed that alpha and delta cells exposed in vitro to IFNγ display a transcriptomic profile very similar to that of beta cells, with an increased expression of inflammation key genes such as MHC class I molecules, the CXCL10 chemokine and the programmed death-ligand 1 (PD-L1), three hallmarks of IFNγ signaling. Interestingly, at low IFNγ concentration, we observed two beta cell populations (responders and non-responders) based on PD-L1 protein expression. Our data indicate that this differential sensitivity relies on the location of the cells within the islet rather than on the existence of two different beta cells subsets. The same findings were corroborated by the in vivo analysis of pancreatic islets from the non-obese diabetic mouse model of T1D, showing more intense PD-L1 staining on endocrine cells close to immune infiltrate. Collectively, our work demonstrates that alpha and delta cells are as sensitive as beta cells to IFNγ, and suggests a gradual diffusion of the cytokine into an islet. These observations provide novel insights into the in situ inflammatory processes occurring in T1D progression.

## 1. Introduction

Type 1 diabetes is a worldwide autoimmune disease characterized by chronic hyperglycemia due to the destruction of insulin-producing pancreatic beta cells. Beta cell death occurs by cell-to-cell contact with autoreactive T lymphocytes [1,2,3] and by contact-independent mechanisms where inflammatory cytokines play a key role [4,5]. The cytotoxic effect of cytokines on whole pancreatic islets [6,7,8] and on beta cells [9,10,11,12,13] is well-known.

Among these cytokines, IFNγ has been detected in pancreatic lymphocytes from human diabetic patients [14]. Its mRNA is expressed by the immune cells infiltrating the islets, and correlates with destructive insulitis in rodent models of diabetes [15,16]. An IFNγ signature has been observed both in human pancreatic endocrine cells from diabetic patients [17] and in islets from the non-obese diabetic (NOD) mouse [18], a model of T1D [19,20]. In mice, antibodies against IFNγ decrease the incidence of cyclophosphamide-induced diabetes [21] and IFNγ-deficient mice are resistant to virus-induced diabetes [22]. Conversely, transgenic mice with beta cell specific IFNγ expression become diabetes prone [23]. IFNγ participates to diabetes progression by increasing beta cell visibility to immune cells through the upregulation of HLA class I molecules [24,25,26] and presentation of beta cell-derived peptides [27,28]. Another key feature of IFNγ is its capacity to induce at the surface of beta cells the expression of programmed death-ligand 1 (PD-L1) [29,30], an immune checkpoint protein [31]. PD-L1 may slow-down beta cell destruction and thus T1D development since its deficiency increases the incidence of diabetes both in NOD mice and in a model of diabetes induced by splenocytes transfer [32]. Additionally it has been suggested that pancreatic expression of PD-L1 delays beta cell destruction [33].

While beta cells are at the centre of diabetes research, they are not the sole endocrine cell type in the pancreas. Indeed, beta cells cluster with glucagon-secreting alpha cells and somatostatin-secreting delta cells to form the endocrine units of the pancreas: the islets of Langerhans. Importantly, alpha and delta cells also show dysregulation during diabetes. For example, impaired glucagon secretion among diabetic patients has been identified for decades [34,35]. This result is supported by recent experiments performed on islets isolated from diabetic patients showing both a decreased expression of key alpha cells-enriched transcription factors, and a loss of glucose-regulated glucagon secretion [36,37]. A limited number of studies also suggest an alteration of delta cells function in type 1 diabetes [38,39]. Yet, information on alpha and delta cell functions and fates in T1D remains scarce. Interestingly, the different endocrine cell types within islet are tightly interacting [40,41,42], making conceivable that beta cell dysfunction or loss imbalances the neighboring alpha and delta cell function. Another hypothesis is that within islets, alpha and delta cells directly suffer from the diabetogenic milieu. Here, we asked whether IFNγ that impacts beta cells, also signals on alpha and delta cells.

To address this question, we treated pancreatic mouse islet with IFNγ, FACS-sorted alpha, beta and delta cells using our recently developed FACS-based approach [43] and performed RNA-seq analysis. We show that alpha, beta and delta cells are surprisingly similar in their response to IFNγ. We also observed two populations of beta cells based on their response to IFNγ, reflecting a spatially restricted effect of IFNγ linked to its gradual diffusion across the islet.

## 2. Materials and Methods

### 2.1. Animal Procedure

All of the animal studies complied with the ARRIVE guidelines and were conducted in strict accordance with the EU Directive 2010/63/EU for animal experiments and with regard to specific national laws and INSERM guidelines. Twelve weeks old C57BL/6JRj male mice were obtained from Janvier Labs (Saint Berthevin, France). NOD mice were bred and housed under specific pathogen-free conditions. Only female NOD mice were used in this study. Mice were maintained on a 12:12 light-dark cycle and provided with water and food ad libitum. They were killed by cervical dislocation. All experiments were approved by the Ethical Committee of Paris Cité University and the French Ministry of Higher Education and Research #16376-2017122210502504.

### 2.2. Isolation, Culture and Treatment of Mouse Pancreatic Islets

Mouse islet isolation was performed as described [43]. Briefly, Type V collagenase (#C9263, Sigma Aldrich, St. Louis, MO, USA) was injected into the common bile duct while the ampulla of Vater was clamped. Inflated pancreas was collected and digested at 37 °C for 20 min. Digested pancreas was resuspended in HBSS (#14025, Thermo Fisher Scientific, Waltham, MA, USA) and islets were then handpicked. Islets were cultured in RPMI 1640 (#61870-010, Thermo Fisher Scientific) containing 10% fetal calf serum (FCS CVFSVF00-01, Eurobio, Les Ulis, France) and penicillin/streptomycin (#15140122, ThermoFisher Scientific). They were treated with IFNγ (#485-MI-100, R&D systems, Minneapolis, MN, USA) for 20 h.

### 2.3. Flow Cytometry

Islets were dispersed in single cell suspensions using the Neural Tissue Dissociation Kit (#130-092-628, Miltenyi Biotec, Bergisch Gladbach, Germany) or using Accutase (#07922, Stemcell Technologies, Vancouver, BC, Canada) for reaggregation experiment. Reaction was stopped by adding 1 mL of HBSS. Cell surface staining was performed as described [43]. Briefly, dispersed islet cells were centrifuged and resuspended in FACS medium (HBSS 10% FCS) with antibodies for 15 min at 4 °C in the dark. Antibodies used were: Epcam (1:100, #118227, Biolegend, San Diego, CA, USA) CD31 (1:100, #102522, Biolegend) CD45 (1:100, #103132, Biolegend) CD24 (1:100, #101840, Biolegend) CD71 (1:100, #113806, Biolegend) CD49f (1:200, #313612, Biolegend) TER119 (1:100, #116228, Biolegend) IFNGR (1:100, #740032, BD Biosciences, San Jose, CA, USA) PD-L1 (1:100, #563369, BD Biosciences) H-2 class I (1:100, #479744, BD Biosciences). Then, cells were rinsed and resuspended in FACS medium with propidium iodide (1/4,000, #P4864, Sigma Aldrich) before FACS acquisition. Epcam antibody was omitted when cells were prepared for reaggregation experiments. For each antibody, optimal dilution was determined by titration. Cell sorting was carried out using a FACSAria III (BD Biosciences). Data were analyzed using FlowJo™ Software 10.6.1 (RRID:SCR_008520, BDBiosciences).

### 2.4. Reaggregation Experiments

2% agarose (#A9539, Sigma Aldrich) was poured into silicone-micro cast (#Z764043-6EA, Sigma Aldrich). After gelling at room temperature, agarose molds were rinsed for 30 min at 37 °C with RPMI1640 containing 10% FCS. Dispersed islet cells or sorted cells (200,000) were seeded in 96-well agarose molds and incubated at 37 °C. Pseudo-islets were harvested by flipping agarose molds upside-down.

### 2.5. RNA Extraction, Reverse Transcription and qPCR

RNA extraction was performed on islets or cells sorted in RLT buffer (#74004, Qiagen, Hilden, Germany) using the RNeasy Micro kit (#74004, Qiagen) according to the manufacturer’s protocol, cDNAs were obtained using Maxima First Strand cDNA synthesis Kit (#K1642, Thermo Fisher Scientific) according to the manufacturer’s protocol. qPCR was performed with Power SYBR Green Master Mix (#4367659 Thermo Fisher Scientific). Primers sequences (Eurofins France, Nantes, France) are presented in Appendix A. qPCR was performed on a QuantStudio 3 (Thermo Fisher Scientific,). The relative expression was calculated according to the 2^−ΔCt^ method using *Ppia* as a housekeeping gene, and fold changes were assessed using the 2^−ΔΔCt^ relative to the control group.

### 2.6. Immunohistochemistry

For immunohistochemistry, islets or whole pancreas were fixed with 3.7% formalin (252549, Merck, Darmstadt, Germany), pre-embedded in agarose (in the case of islets) (#A4018, Sigma Aldrich), embedded in paraffin (Histowax, Histolab, Copenhagen, Denmark), and sliced in 4-μm-thick sections (#RM2145, Leica, Nussloch, Germany). They were deparaffinized with xylene and ethanol washes and boiled in the antigen retrieval solution (#HK086-9K, BioGenex, Fremont, CA, USA) using a pressure cooker (#DC2012-220V, Biocare Medical, Pacheco, CA, USA). They were then permeabilized and blocked with a solution of TBS (#T6664-10PAK Sigma Aldrich), BSA 3% (#A7906, Sigma Aldrich) and Triton 0.3% (#T8787, Sigma Aldrich) for 30min at RT. Antibodies were diluted in the same solution. Antibodies used were PD-L1 (1:50, AF1019, Bio-techne, Minneapolis, MN, USA) PDX1 (1:1000, [44]) Anti goat (1:400, A-21467, ThermoFisher) Anti Rabbit (1:200, 711-076-152, Jackson ImmunoResearch, West Grove, PA, USA). Primary antibodies were incubated overnight at 4 °C, and secondary antibodies for 3 h at RT.

### 2.7. Next Generation Sequencing and Bioinformatics

RNA was isolated using the RNeasy Micro kit (#74004, Qiagen) according to the manufacturer’s protocol. RNA quality was verified by electrophoresis using an Agilent 2100 Bioanalyzer (Agilent Technologies, Palo Alto, CA, USA). For library construction, 5 ng of high quality (RIN > 7) total RNA was processed using an Ovation Solo RNA-seq Kit (#0501-96, NuGEN, Leek, The Netherlands) per the manufacturer’s instructions. Briefly, total RNA was treated with DNase I and reverse-transcribed using random primers. Chemical treatment during second strand synthesis enabled us to achieve strand specificity. After end repair, adaptor ligation, and library amplification, depletion of rRNA was realized using AnyDeplete (NuGEN). Libraries were then quantified with a Qubit HS DNA assay (#Q32855, Thermo Fisher Scientific) and library profiles were assessed using a DNA High Sensitivity LabChip Kit on an Agilent 2100 Bioanalyzer. Libraries were sequenced on an Illumina Nextseq 500 instrument (San Diego, CA, USA) using 75 base lengths reading V2 chemistry in a paired-end mode. After sequencing, a primary analysis based on AOZAN software (Genomic Paris Centre, Ecole Normale Supérieure, Paris, France) was applied to demultiplex and control the quality of the raw data (based on FastQC modules version 0.11.5). Sequencing reads were mapped to the mouse genome version GenCode M23 (RRID: SCR_014966 and GRCm38.p6 release 93) using STAR v2.5.2b [45] (RRID: SCR_015899). Transcripts were quantified using the RSEM software tool [46] (RRID: SCR_013027) Normalization and differential gene expression DGE analyses were performed with R package DESeq2 v1.32.0 (10.1186/s13059-014-0550-8), including batch in the design formula, and Benjamini-Hochberg correction was applied with threshold for significance set at adjusted *p*-values < 0.05. Figures were plotted with R packages ComplexHeatmap v2.8.0 [47].

### 2.8. Statistical Analyses

Each *n* represents an independent biological sample. All of the graphs show means ± Standard Deviation (SD). Statistical analysis was conducted using Prism v5.0 (GraphPad Software, San Diego, CA, USA) applying an unpaired *t* test. A *p* value < 0.05 was considered statistically significant.

## 3. Results

### 3.1. Mouse Alpha, Beta, and Delta Cells Respond to IFNγ in a Similar Manner

We investigated the response of mouse islet endocrine cell subsets to IFNγ. We isolated pancreatic islets from C57BL/6 mice, treated them for 20 h with IFNγ (5 ng.mL^−1^), FACS-sorted alpha, beta and delta cell populations and performed RNA-seq (Figure 1a). The sorting profile of alpha, beta and delta cell populations using our previously described panel of antibodies (CD24, CD49f and CD71) [43] was not altered following IFNγ treatment (compare Appendix A). Moreover, purity of alpha, beta and delta cell populations analyzed by RT-qPCR was not perturbed by IFNγ treatment: neither hormone enrichment (*Ins1*, *Gcg*, *Sst*) nor duct (*Spp1*) or acinar (*Amy1*) contaminations were modified by IFNγ treatment (Appendix A). RNA-seq data analyses indicated that exposure to IFNγ modulated the expression of many genes in each cell subset. The majority of regulated genes were induced and very few repressed (160 vs. 24 in alpha cells; 174 vs. 0 in beta cells, and 204 vs. 29 in delta cells; adjusted *p*-value < 0.01). We next extracted genes whose expression was induced by IFNγ in alpha cell population and asked whether they were also induced in beta or delta cells. We performed the same analysis by extracting genes induced in beta or delta cells. We observed a significant overlap of the genes induced by IFNγ in the three endocrine cell populations (Figure 1b). The distribution of those genes among alpha, beta and delta cells are presented in (Appendix A).

Finally, we selected in each endocrine population, based on adjusted *p*-values, the top 50 differentially expressed genes induced by IFNγ. We obtained a list of 66 unique genes. Many of them (48/66) top ranked in at least two populations (highlighted with red dots) and the remaining ones were also significantly induced in the other cell types although not ranked in the top 50 (Figure 1c). We validated RNA-seq data by RT-qPCR analyzing *Cd274* mRNA encoding for PD-L1 and *H2*(*dlb1q1*) mRNA encoding for the alpha chain of the major histocompatibility complex (H-2 class I), two previously validated IFNγ targets in pancreatic beta cells [29,48]. Their inductions were proportional to IFNγ concentrations in whole islets and in sorted alpha, beta and delta cell populations (Figure 1d,e).

We next verified at the protein level the effect of IFNγ on the surface expression of PD-L1 and H-2 class I. There, islets were cultured for 20 h with three concentrations of IFNγ (50, 500 and 5000 pg.mL^−1^), and flow cytometry was used to compare and quantify membrane expression of PD-L1 and H-2 class I on alpha, beta and delta cell populations. We observed a similar induction of both proteins in the three endocrine cell subsets (Figure 2a,b for quantification and Appendix A).

Those experiments demonstrate that the three main pancreatic endocrine populations (alpha, beta and delta cells) are sensitive and respond in a similar fashion to IFNγ.

### 3.2. Beta Cell Heterogeneity in Response to IFNγ

Flow cytometry indicated that at high concentration (5 ng.mL^−1^), IFNγ activated cell surface expression of PD-L1 in nearly every alpha, beta and delta cell (Figure 2a). On the other hand, at lower concentrations (50 pg.mL^−1^), we observed that IFNγ turned on PD-L1 surface expression in only a beta cell fraction, the remaining beta cells being refractory to PD-L1 induction (Figure 2a). We obtained similar data when measuring HLA surface expression (Appendix A). We next treated islets for 20 h with 50 pg.mL^−1^ IFNγ and separated by FACS PD-L1^high^ and PD-L1^low^ beta cell populations. We also sorted control beta cells cultured in the absence of IFNγ (Figure 3a). RNA-seq analyses indicated that upon IFNγ exposure, the PD-L1^high^ population had activated an IFNγ signature, which was far less the case in the PD-L1^low^ population (Figure 3b). RT-qPCR data further validated this point. Indeed, IFNγ targets such as *Igtp*, *Irgm2*, *Cxcl10* and *Irf1* were far more induced by IFNγ in PD-L1^high^ as compared to PD-L1^low^ beta cells. (Figure 3c).

To determine whether differences in sensitivity to IFNγ between PD-L1^high^ and PD-L1^low^ beta cells was due to a differential expression of the IFNγ receptor (IFNGR), we performed qPCR for *Ifngr1* and *Ifngr2*. Expression levels were similar in PD-L1^high^, PD-L1^low^ and control beta cells (Figure 4a). Flow cytometry analysis using anti-IFNGR antibodies further supported the fact that differences in IFNγ sensitivity between PD-L1^high^ and PD-L1^low^ beta cell populations was not caused by a differential IFNGR expression (Figure 4b,c for quantification).

It has been reported that less differentiated beta cells exist in the NOD mouse pancreas, that better survive the autoimmune attack [49]. We thus tested the hypothesis that the beta cell population less sensitive to IFNγ would also be less differentiated when compared to IFNγ-sensitive beta cells. We analyzed the expression of genes encoding beta cell specific transcription factors such as *Pdx1*, *MafA*, *Nkx6.1*, Insulin (*Ins1* and *Ins2*), proteins implicated in insulin processing and secretion such as *Pcsk1*, *Slc2a2*, *Ucn3*, *Chga*, protein targets of autoantibodies found in the blood of T1D patients, such as *Slc30a8*, *Ptprn*, *Gad1* and *Cfap126*, a recently described beta cell heterogeneity marker [50]. Overall, no difference could be identified between IFNγ sensitive and insensitive beta cells (Figure 5a).

Functional analyses highlighted different beta cells subsets, referred as heterogeneity [50,51,52,53,54,55]. However, whether beta cell heterogeneity reflects fixed subpopulations or dynamic interchangeable states, remains an opened question [56]. Here, we asked whether PD-L1^high^ and PD-L1^low^ beta cells represent stable populations. We treated mouse islets with IFNγ (50 pg.mL^−1^) for 20h and FACS-purified PD-L1^low^ cells. We reaggregated them for 72 h in culture and treated them a second time during 20 h without or with IFNγ (50 pg.mL^−1^) (Figure 5b). Flow cytometry analyses indicated that about half of the beta cells that did not respond to IFNγ during the first round of treatment, responded during the second round (Figure 5c,d for quantification), a proportion comparable to the one observed during the first incubation on intact islets. Taken together, in islets, some beta cells are insensitive to IFNγ at a given time, however, they can switch to responsive-cells when re-exposed to IFNγ.

### 3.3. Beta Cells That Respond to IFNγ Are Located at the Islet Periphery

It was recently observed that inflammatory cytokines do not freely diffuse between cells, creating gradients of signaling [57]. As islets are micro-organs containing 1000–3000 cells, we tested the hypothesis that IFNγ does not freely circulate into islets. We first tested the effect of IFNγ treatment on dissociated islets. Dispersed islet cells were exposed or not to low concentration of IFNγ (50 pg.mL^−1^), and PD-L1 activation in beta cells was analyzed by flow cytometry. In that experiment setting, IFNγ activated PD-L1 in nearly every beta cell (90.1 ± 5.5%) (Figure 6a,b for quantification). This result suggested that in this dissociation-treatment-reaggregation protocol, IFNγ reached every beta cell, which is not the case in whole islets. We thus used PD-L1 staining on islet sections as a probe to test whether beta cells that respond to IFNγ had a specific location within the islets.

As expected, no PD-L1 staining was detected in control cells while almost all cells were PD-L1 positive when treated with 5 ng.mL^−1^ of IFNγ for 20 h (Figure 6c, compare left to right panels). Interestingly at 50 pg.mL^−1^ of IFNγ, PD-L1^high^ cells were located at the periphery of the islets (Figure 6c middle panel). Quantification of PD-L1 staining intensity in relation to the distance from the islet centre further supports this claim (Figure 6d,e). Thus, the responsiveness of cells to IFNγ relies on their position within an islet.

In NOD mice, T lymphocytes that produce IFNγ first surround the pancreatic islets (peripheral insulitis), and then progressively invade them (invasive insulitis). In an attempt to translate our in vitro observations to an in vivo situation, we analyzed PD-L1 expression on pancreas sections from NOD mice at different time-points. As expected, in young mice (3–6 weeks old), islets were not infiltrated and did not express PD-L1 (Figure 7a, left panels). In older mice (11- and 14-weeks old), peri-insulitis was clearly observed around islets, along with PD-L1 staining within the islets (Figure 7a right panels). Closer examination indicated that beta cells located at proximity of the immune infiltrate expressed the highest PD-L1 levels (Figure 7a–c for quantification).

These experiments show that IFNγ does not freely diffuse within an islet.

## 4. Discussion

Cytokines such as IFNγ are important mediators of inflammation during T1D progression [21,22,58,59]. As an example, an IFNγ signature is observed in islets from NOD mice as early as 6 weeks of age [18]. Transcriptomic changes induced by cytokines were also deeply studied in primary beta cells or beta cell lines [59,60,61,62,63]. However, our knowledge of IFNγ responses in other islet cell populations (glucagon-producing alpha cells and somatostatin-secreting delta cells) is more limited. In the present study, using our recently described FACS-based strategy to sort the major endocrine populations from mouse islets [43], we investigated the effects of IFNγ on mouse alpha and delta cells, and compare to beta cells. Our work pointed out two major findings: (i) alpha, beta and delta cells responses to IFNγ are similar and (ii) the diffusion of IFNγ within an islet is impeded, thus creating a gradient of concentration across the islet.

The current study has been made possible by the development in the laboratory of a FACS-based strategy to purify live alpha, beta and delta cell populations [43]. This approach opens new avenues for pancreatic islets study. As an example, RNA-seq on bulk alpha, beta and delta populations solves both the requirement of complex reporter mice [40,64,65] and the eventual pitfalls of single-cell RNA sequencing [66,67], the main technic used for alpha and delta cells transcriptomic analyses [68,69,70]. The present protocol was set up on untreated mouse islets and we now demonstrate that this method is also robust on islets treated with IFNγ. As described here, this strategy allows the sorting of cells on multiple parameters. Here, we sorted beta cells based on PD-L1 expression following IFNγ treatment. PD-L1 is a stress response factor known to be induced by cytokines such as IFNγ [71], oncogenes such as epidermal growth factor [72] and hypoxia inducible-factor-1 [73]. Sorting was followed by reaggregation into pseudo-islets for further analyses. Such protocol permits to address the question of cell subtype stability, a current topic of major interest [56].

We observed here that IFNγ transcriptomic signature of sorted mouse primary beta cells was similar to previous observations on human beta cells [74], demonstrating the strength of our strategy. Confident with our protocol, we also characterized alpha and delta cells responses to IFNγ. Remarkably, transcriptomic responses to IFNγ were similar between alpha, beta and delta cells, pointing out that beta cells are not the only source of inflammatory response genes in the islet. In this context, gene products activated by IFNγ in alpha and delta cells might impact diabetes development. This might be the case for *Cxcl10* that encodes a chemokine expressed by beta cells, that attracts T lymphocytes and thus participates to diabetes onset and progression [75,76]. We showed here that alpha and delta cells also increase *Cxcl10* expression upon IFNγ exposure. Recently, alpha cells expressing the CXCL10 protein were reported both in diabetic mouse model and in human islets from diabetic patients [77]. Our data suggest that this induction of CXCL10 relies on IFNγ, and add the fact that delta cells also contribute to CXCL10 production within islets. Thus, alpha and delta cell may favor, along with beta cells, infiltration of the pancreatic islets by immune cells. PD-L1 also represents another example of proteins upregulated by IFNγ in alpha and delta cells that might impact diabetes development. It has been shown that inflamed islets cells may limit immune cells activation through PD-L1/PD1 interaction [32,33]. With our experimental model, we confirmed that beta cells express high levels of PD-L1 in inflammatory condition as previously observed [29,49,78]. Interestingly, we demonstrate here that alpha and delta cells also increased *Cd274* mRNA, leading to an increase of PD-L1 surface expression confirmed by flow cytometry. Whether PD-L1 expressed by alpha and delta can slow down the immune attack remains an open question.

It is well documented that beside inducing gene expression, a combination of cytokines—including IFNγ—precipitates beta cell death [10,79]. A limited number of studies performed on alpha cell lines suggest that cytokines can also induce alpha cell apoptosis, but to a lesser extent than beta cells. [80]. It was recently hypothesized that the alpha cell resistance to cytokine-induced apoptosis relies on their high expression of poly-ADP Ribose Polymerase PARP-14, that modulates the expression of apoptosis-related genes [81]. In our dataset, using IFNγ, we indeed observed an increased expression of *Parp14* in alpha cells exposed to IFNγ, but similar to beta and delta cells. So, our results do not support an alpha cell resistance to IFNγ based on PARP-14.

Another major point derived from our study concerns differences in individual beta cell sensitivity to IFNγ within islets. Heterogeneity among beta cells has been identified for several decades. Data published in the 80th and 90th and reviewed in a landmark publication [82] indicated functional beta cell heterogeneity. As an example, glucose stimulates proinsulin biosynthesis in a dose-dependent manner by recruiting additional beta cells, that might suggest differences in beta cell sensitivity to glucose [83]. Moreover, in vivo, in rats, beta cells showed a regional heterogeneity in terms of their ability to respond to stimulation by insulin secretagogues [51]. In the present study, by measuring cell surface levels of PD-L1 following low-dose IFNγ stimulation, we observed two beta cell populations: PD-L1^high^ cells were IFNγ-responsive while the PD-L1^low^ cells were not, without difference in expression of surface IFNGR. PD-L1^high^ and PD-L1^low^ populations were previously described in other cell types such as a fibroblast cells line, but also murine lung, liver and kidney where they were linked to senescence [84] Heterogeneity in PD-L1 expression was also previously described in NOD mice where a population of beta cells that resists auto-immune attack and express high level of PD-L1 was described [49]. Interestingly, in that model, cells expressing higher levels of PD-L1 seemed to be less differentiated and expressed lower levels of beta cell markers such as *Ins1*, *Ins2*, *Mafa*, *Pdx1*, *Nkx6.1*, *Slc2a2* while re-expressing markers of pancreatic endocrine progenitors such as *Ngn3* [49]. However, in our model, we did not observe any difference in the expression of such genes, indicating similar differentiation status between PD-L1^high^ and PD-L1^low^ beta cells.

We also assessed whether PD-L1^high^ and PD-L1^low^ beta cells represent stable populations fixed in a specific state or whether cells might navigate from one state to the other. Recent data based on sorting of beta cell populations [50,53] or single cell transcriptomic analyses [68,69,70,85,86] suggested that beta cells are fixed in transcriptional and functional states, giving rise to stable and specific beta cell populations entitled Beta-1, -2, but this remains a matter of discussion. As examples, cell tracking [87] and pseudo-time trajectory [88] support the notion that cells swap between different states. Our data indicate that PD-L1^high^ and PD-L1^low^ beta cells represent transient populations. Indeed, PD-L1^low^ beta cells were able to induce PD-L1 surface expression in response to a second pulse of IFNγ. Moreover, IFNγ stimulation of dispersed islet cells resulted in all beta cells being PD-L1^high^, further suggesting that the location of a beta cell within an islet might dictate its sensitivity to IFNγ. Interestingly, tissular diffusion of IFNγ was recently addressed with melanoma cells as the penetrance of the cytokine was dependent on the density of cytokine-consuming cells in the tissue, creating a gradient of concentration [57,89]. Our data derived from pancreatic islets treated in vitro with IFNγ and from pancreases of NOD mice suggest that a similar gradient mechanism might contribute to our observations, pancreatic islet being a cell-dense structure. There, IFNγ would be captured by the first layer of cells, protecting the cells located inside the islet. Another eventual actor of this process might be the basal membrane surrounding the islet, which has been shown to protect islets from immune invasion [90]. The basal membrane might limit the diffusion of soluble molecules such as cytokines, as suggested by the observation that transplanted islets show better resistance to cytokines when components of extracellular matrix are added [91,92].

Taken together, our data, suggest that we should re-consider the importance of endocrine non-beta cells in T1D. Indeed, alpha and delta cells are similarly affected by inflammation and respond with induction of genes that may influence the recruitment and activation of autoreactive T cells, and thus disease progression, such as *Cxcl10* and *Cd274*. Our work also proposes a new insight into the propagation of inflammation across an islet, where an islet would shield itself against inflammatory stimuli such as IFNγ entry. Experiments focusing on islet inflammation should take this notion into account. In this study, experiments were performed using mouse islets. Future studied will aim at determining whether similar results can be obtained with either pseudo-islets made of homogeneous human beta cell lines or with human islets. It will also be interesting to test whether other cytokines (Il-1β, TNFα) show similar gradient-like action.

## Figures and Tables

**Figure 1 cells-12-00113-f001:**
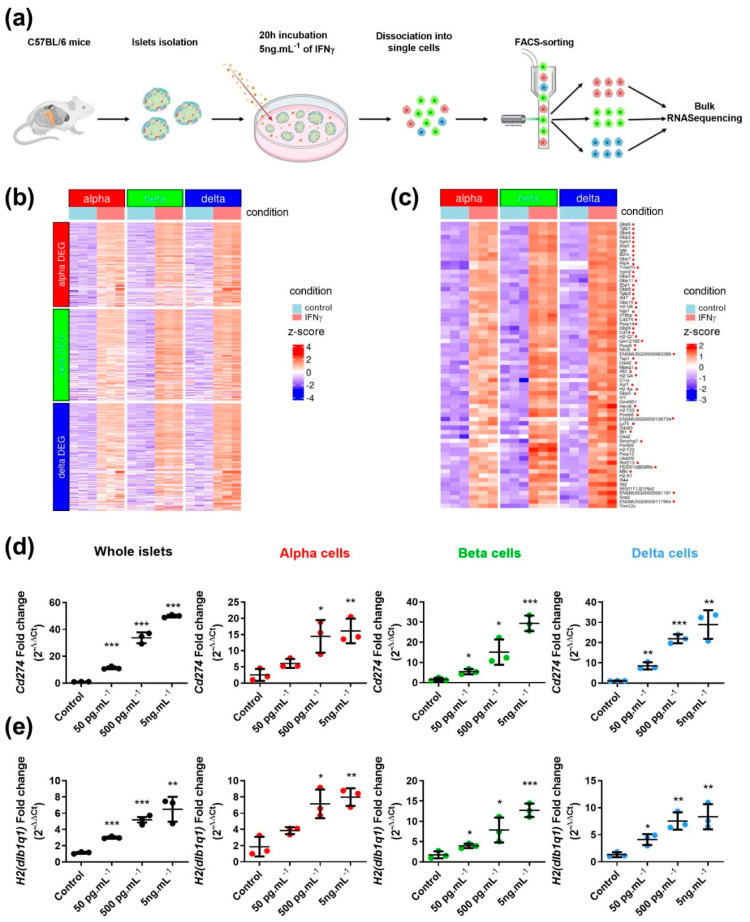
Mouse alpha, beta, and delta cells respond to IFNγ in a similar manner at transcriptional level. (**a**) Schematic protocol. (**b**) Heatmap presenting genes induced by IFNγ in alpha, beta and delta cells (ajusted *p*-value < 0.01) (*n* > 3 independent experiments). (**c**) Heatmap presenting the top 50 genes induced by IFNγ in alpha, beta and delta cells, resulting in a list of 66 unique genes. Genes are indicated with a red dot when they top-ranked in at least two populations. (**d**,**e**) Expression of *Cd274* (**d**) and *H2*(*dlb1q1*) (**e**) mRNAs measured by RT-qPCR on whole islets, purified alpha, beta and delta cells following islets treatments with different concentration of IFNγ (0–5 ng.mL^−1^) for 20 h (*n* = 3 independent experiments). Data are represented as mean ± SD, * *p* < 0.05, ** *p* < 0.01, *** *p* < 0.001.

**Figure 2 cells-12-00113-f002:**
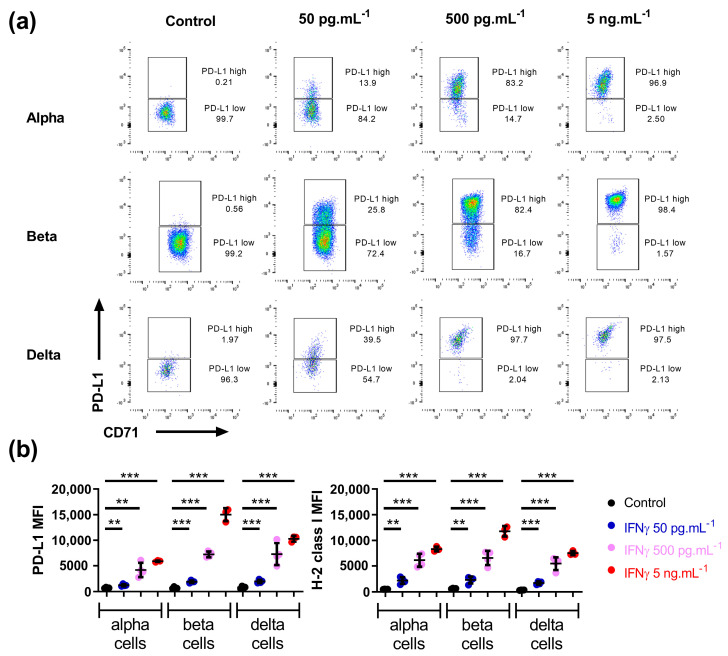
Mouse alpha, beta, and delta cells respond to IFNγ in a similar manner at protein level. (**a**) Representative flow cytometry plots showing PD-L1 surface expression on alpha, beta and delta cells from islets treated with different concentration of IFNγ (0–5 ng.mL^−1^) for 20 h. (**b**) Mean Fluorescence Intensity (MFI) of PD-L1 (**left**) and H-2 class I (**right**) in alpha, beta and delta cells after treatment of islets with different concentration of IFNγ (0–5 ng.mL^−1^) for 20 h (*n* = 3 independent experiments). Data are represented as mean ± SD, ** *p* < 0.01, *** *p* < 0.001.

**Figure 3 cells-12-00113-f003:**
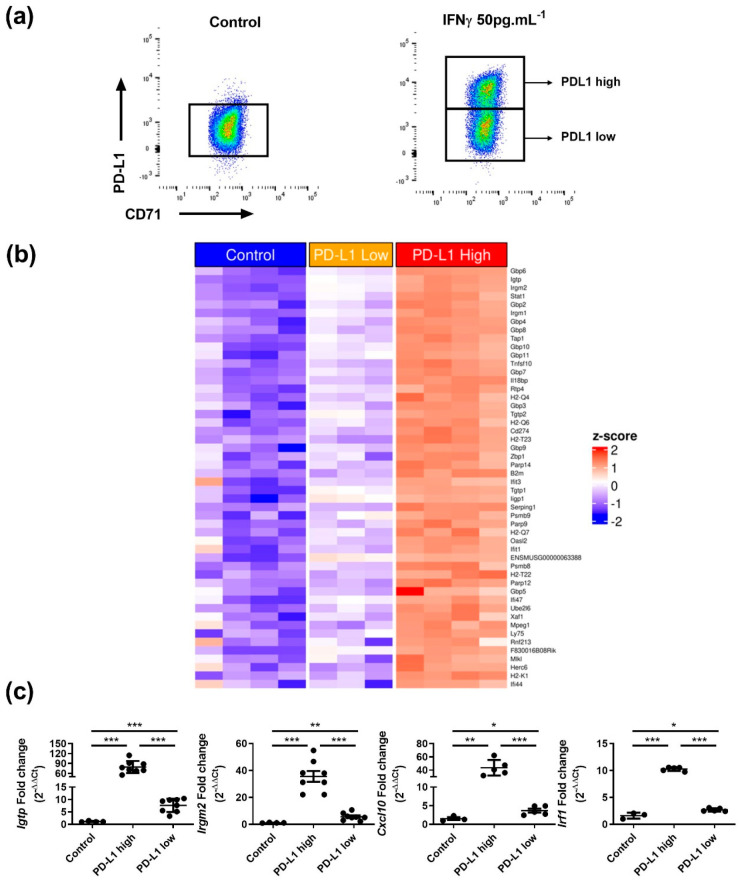
Beta cell heterogeneity in response to IFNγ. (**a**) Representative flow cytometry plots of PD-L1 surface expression on beta cells from control or after treatment with 50 pg.mL^−1^ of IFNγ for 20h. (**b**) Heatmap presenting the top 50 IFNγ induced genes in beta cells in control, PD-L1^low^ or PD-L1^high^ beta cells (*n* = 3 independent experiments). (**c**) Expression measured by RT-qPCR of *Igtp*, *Irgm2*, *Cxcl10* and *Irf1* mRNAs in control, PD-L1^high^ and PD-L1^low^ beta cells. (*n* = 3–8, from 3–4 independent experiments). Data are represented as mean ± SD, * *p* < 0.05, ** *p* < 0.01, *** *p* < 0.001.

**Figure 4 cells-12-00113-f004:**
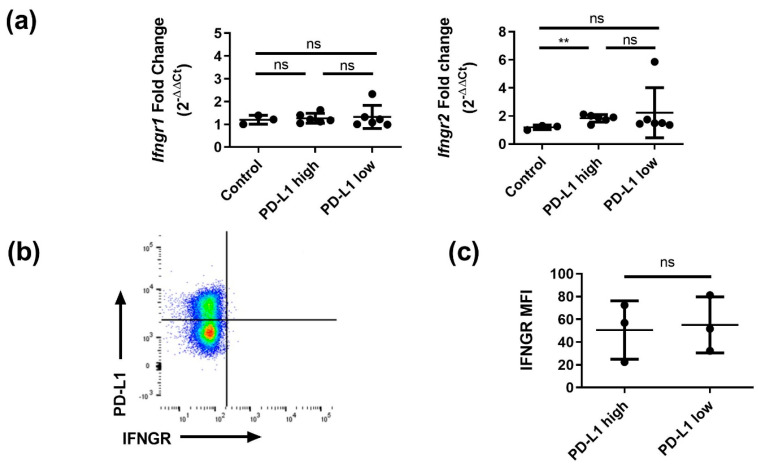
Similar IFNγ receptor expression in PD-L1^high^ and PD-L1^low^ beta cells. (**a**) Expression measured by RT-qPCR of *Ifngr1* and *Ifngr2* mRNAs in control, PD-L1^high^ and PD-L1^low^ beta cells (*n* = 3–6, from 3 independent experiments). (**b**) Representative flow cytometry plot of PD-L1 and IFNGR surface expression on beta cells from islets treated with 50 pg.mL^−1^ of IFNγ for 20h. (**c**) Scatter plot showing IFNGR MFI in PD-L1^high^ and PD-L1^low^ beta cells (*n* = 3 independent experiments). Data are represented as mean ± SD. ** *p* < 0.01, ns: not significant.

**Figure 5 cells-12-00113-f005:**
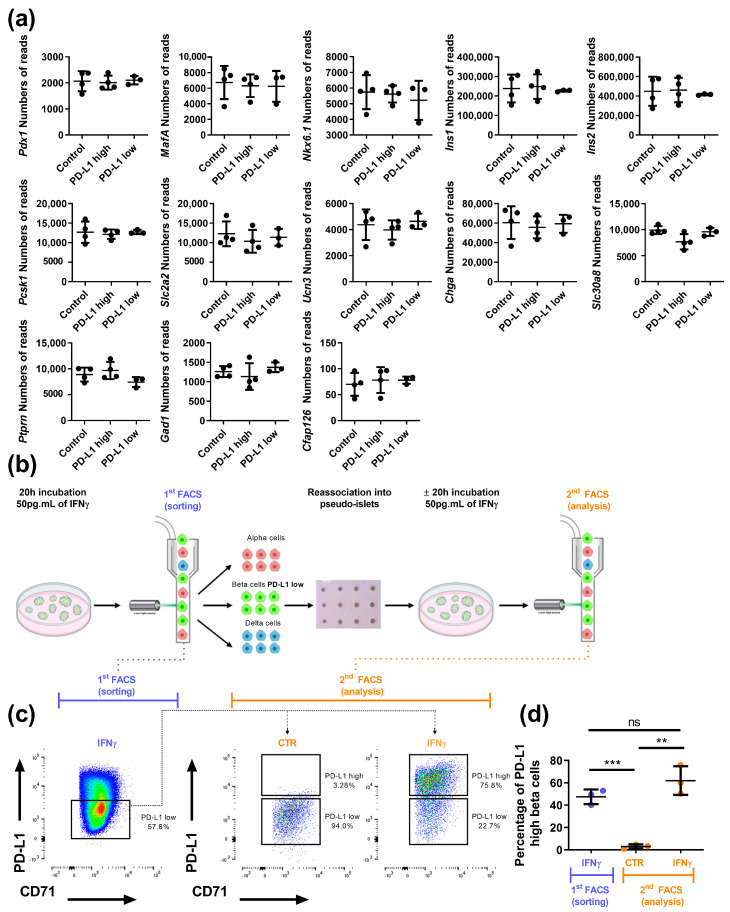
PD-L1 is inducible in the PD-L1^low^ population. (**a**) Gene expression from the RNA-sequencing data in control, PD-L1^high^ and PD-L1^low^ beta cells (*n* = 3 independent experiments). (**b**) Schematic protocol to test PD-L1 induction in the PD-L1^low^ beta cell population. (**c**) Representative flow cytometry plots of PD-L1 surface expression on beta cells from islets treated with 50 pg.mL^−1^ IFNγ for 20 h (**left**). PD-L1^low^ beta cells, along with alpha and delta cells, were sorted and reaggregated for 3 days. On the 4th day, pseudo-islets were harvested, incubated with or without 50 pg.mL^−1^ of IFNγ for 20 h and analyzed by FACS for PD-L1 surface expression. (**d**) Percentage of PD-L1^high^ beta cells on the 1st FACS (islets treated with 50 pg.mL^−1^ of IFNγ for 20 h) and on the 2nd FACS (PD-L1^low^ pseudo-islets cultured in control medium or with a second pulse of IFNγ, 50 pg.mL^−1^ for 20 h) (*n* = 3 independent experiments). Data are represented as mean ± SD, ** *p* < 0.01, *** *p* < 0.001.

**Figure 6 cells-12-00113-f006:**
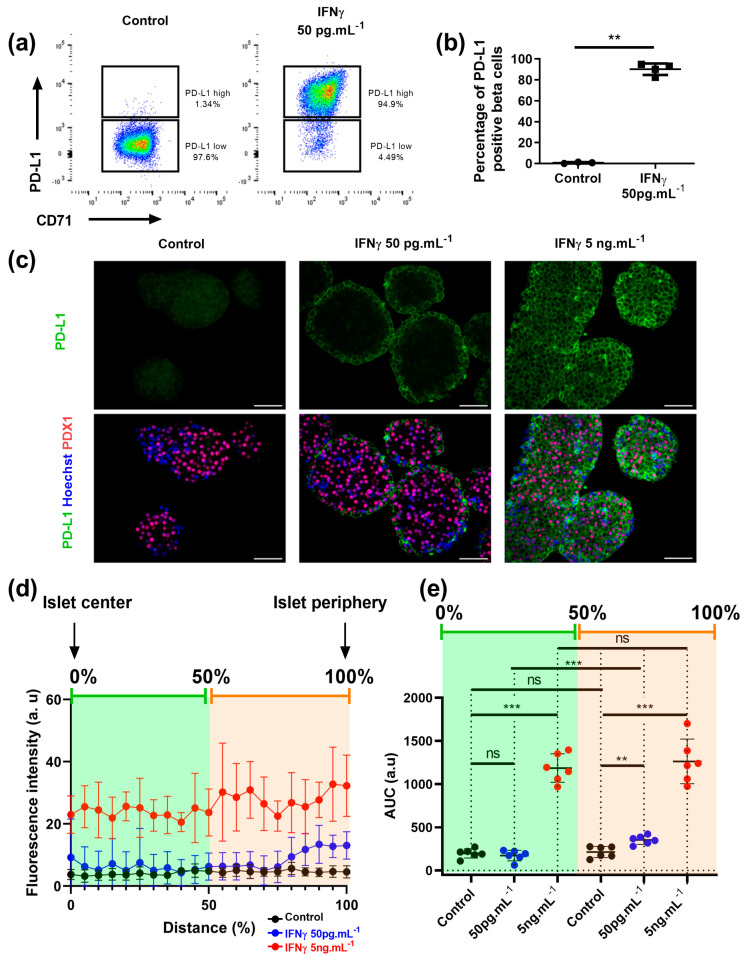
Beta cells that respond to IFNγ are located at the islet periphery (**a**,**b**) Dissociated islet cells were treated with or without 50 pg.mL^−1^ IFNγ and seeded in agarose molds for 20 h, followed by FACS acquisition. (**a**) Representative flow cytometry plots of PD-L1 surface expression; (**b**) Frequency of PD-L1^high^ beta cells (*n* = 3 independent experiments). (**c**–**e**) Islets were treated for 20 h without or with 50 pg.mL^−1^ or 5 ng.mL^−1^ IFNγ, sectioned, and stained for PD-L1 and PDX1. (**c**) representative stainings, Bar: 50 μm. (**d**,**e**) Quantification of PD-L1 intensity in 6 islets from 3 independent experiments. (**d**) Fluorescence intensity is presented in arbitrary units, in function of the distance from the centre of the islet. (**e**) Area under the curves from (**d**) is shown (in green, centre of the islet; in orange, islet periphery). Data are represented as mean ± SD, ** *p* < 0.01, *** *p* < 0.001.

**Figure 7 cells-12-00113-f007:**
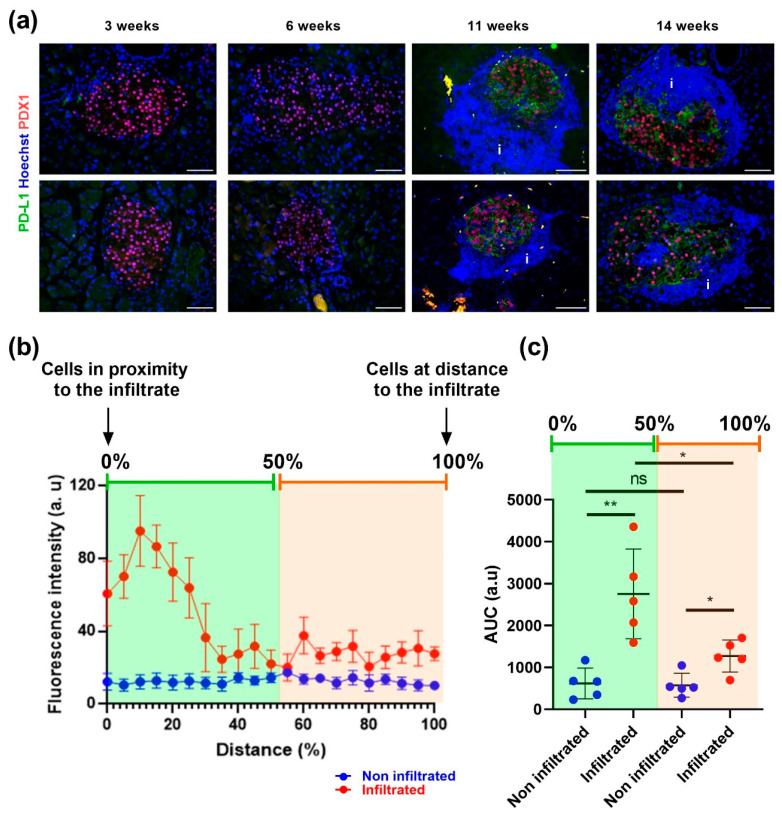
In NOD mice, beta cells located at proximity to the infiltrate express the highest PD-L1 levels Pancreatic sections from NOD mice at different ages (3, 6, 11 and 14 weeks-old) were double stained for PD-L1 (green) and PDX1 (red). (**a**) Representative staining for PD-L1 (green) and PDX1 (red). “i” indicates the immune infiltrate. Bar: 50 μm. (**b**,**c**) Quantification of PD-L1 intensity in 5 independent islets from either a 14 weeks-old NOD mouse that did not present any lymphocyte and was used as baseline versus infiltrated islets from three NOD mice (11–14 weeks-old). (**b**) Fluorescence intensity is presented in arbitrary units, relative to the infiltrate distance. (**c**) Area under the curves from (**b**) is shown (in green, in proximity to the infiltrate; in orange, at distance to the infiltrate). Data are represented as mean ± SD, * *p* < 0.05, ** *p* < 0.01.

## Data Availability

The data presented in this study are openly available in the NCBI’s Gene Expression Omnibus (GEO) (accession GSE221081 and GSE221078).

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
