# Peer review of "Pancreatic Islet Cells Response to IFNγ Relies on Their Spatial Location within an Islet"

_cells, 2022, doi:10.3390/cells12010113_

Round 1

Reviewer 1 Report

This is a great manuscript and given the importance of PD-1/PDL-1 it is also timely.  The experimental work presented here is of high quality and I have no specific recommendations. I would suggest the authors broader the discussion to talk about PDL-1 upregulation as a stress response and also discuss the recent idea presented in this paper https://rdcu.be/c02B0.

Author Response

Reviewer 1

This is a great manuscript and given the importance of PD-1/PDL-1 it is also timely.  The experimental work presented here is of high quality and I have no specific recommendations. I would suggest the authors broader the discussion to talk about PDL-1 upregulation as a stress response and also discuss the recent idea presented in this paper https://rdcu.be/c02B0.

We thank the reviewer for starting its comment with the sentence: “This is a great manuscript”.

We further discussed PD-L1 upregulation as a stress response on page 13. We added the following text ‘’Here, we sorted beta cells based on PD-L1 expression following IFNγ treatment. PD-L1 is a stress response factor known to be induced by cytokines such as IFNγ (70), oncogenes such as epidermal growth factor receptor (71) and hypoxia inducible-factor-1-alpha (72).

We also discussed the recent idea presented in https://rdcu.be/c02B0. on page 14. We added the following sentence ‘’ PD-L1high and PD-L1low cell populations were previously described in other cell types such as fibroblast cell line, but also murine lung, liver and kidney where they were linked to senescence (86).

Reviewer 2 Report

The study by De Burhgrave et al illustrates the role for the spatial location of the different endocrine cells within the islets rather than a cellular heterogeneity in their response to IFNg stimulation. The conclusions are supported by a very elegant set of experiments conducted on handpicked mouse islets and FACS sorted endocrine cells. The immunohistochemistry experiments performed on NOD mice at different ages validate a differential expression of PDL1 according to the cell localization during diabetes development and the notion of cytokine gradient. The study is very interesting and relevant to understand the amplification of the inflammation in T1D.

Yet, the report seems to contrast with earlier work describing a cellular heterogeneity, mainly based on a different differentiated stage, within the islets (Dorrell C et al 2014 ref52; van der Meulen T et al 2017 ref40…). It would be important to extend the characterization presented in Figure 5A to show expression of Fltp, GLUT2 and UCN3 in PDL1 low/high cells.

To exclude further the participation of cellular heterogeneity to the gradual IFNg response observed, the authors may consider evaluating PDL1 expression in a dose response cytokine stimulation on beta cell pseudoislets made of homogeneous beta cells (e.g. min6 or EndoCBH1). 

Finally, the study will be strengthened by validation in human islets and dispersed primary human cells.

Minor points:

On figure 1, the authors are presenting the transcriptomic profiles of alpha, beta and delta cells. In this figure it would be interesting to include Venn diagrams to highlight overlapping genes as well as cell specific genes modulated by IFNg treatment.

Author Response

Reviewer 2

The study by De Burhgrave et al illustrates the role for the spatial location of the different endocrine cells within the islets rather than a cellular heterogeneity in their response to IFNg stimulation. The conclusions are supported by a very elegant set of experiments conducted on handpicked mouse islets and FACS sorted endocrine cells. The immunohistochemistry experiments performed on NOD mice at different ages validate a differential expression of PDL1 according to the cell localization during diabetes development and the notion of cytokine gradient. The study is very interesting and relevant to understand the amplification of the inflammation in T1D.

We thank the reviewer for its highly positive comment

Yet, the report seems to contrast with earlier work describing a cellular heterogeneity, mainly based on a different differentiated stage, within the islets (Dorrell C et al 2014 ref52; van der Meulen T et al 2017 ref40…). It would be important to extend the characterization presented in Figure 5A to show expression of Fltp, GLUT2 and UCN3 in PDL1 low/high cells.

Fltp (Cfap126) expression is now added on Fig 5 and in the text on page 7 with a reference. We added the following sentence ‘’ Cfap126, a recently described beta cell heterogeneity marker (49)’’. Glut2 (encoded by the gene Scl2a2) and Ucn3 were already presented in PDL1 low/high cells

To exclude further the participation of cellular heterogeneity to the gradual IFNg response observed, the authors may consider evaluating PDL1 expression in a dose response cytokine stimulation on beta cell pseudoislets made of homogeneous beta cells (e.g. min6 or EndoCBH1). Finally, the study will be strengthened by validation in human islets and dispersed primary human cells.

We agree that the two sets of experiments proposed by the reviewer would further increase the strength of the manuscript. However, to be properly performed they would take a number of months. We have thus decided to propose such experiments in the discussion section as future experiments. We added the following sentence on page 14: “In this study, experiments were performed using mouse islets. Future studies will aim at determining whether similar results can be obtained with either pseudo-islets made of homogeneous human beta cell lines or with human islets”.

Minor points:

On figure 1, the authors are presenting the transcriptomic profiles of alpha, beta and delta cells. In this figure it would be interesting to include Venn diagrams to highlight overlapping genes as well as cell specific genes modulated by IFNg treatment.

We have now added a Venn diagram on Figure S1D as suggested by the reviewer

Reviewer 3 Report

In the study by De Burghgrave et al., the authors performed incubations of islets in vitro with IFN-g and show that all cell types within the islet respond with similar gene expression patterns and that among beta cells, those that are situated in the periphery are the ones most likely to respond to IFN-g.  These data are consistent with the notion that diffusability of compounds and proteins does not occur freely or uniformly within the islet.  The authors correlate these findings in vitro to findings from pancreas sections from NOD mice, wherein expression of PD-L1 is observed in regions adjacent to insulitis.  Overall, the study appears well performed and designed, and the interpretation of the data is appropriate.  However, messaging in this study is not unexpected or particularly new, unless the authors were to show that the properties of diffusion are different or unique for IFN-g (compared to another cytokine, e.g. IL-1b). The comment at the end of the results section does not seem to be supported by the data in this study (“the notion of beta cell heterogeneity should be taken with care”)—it is not clear why the authors are raising a doubt regarding heterogeneity, simply because all beta cells are capable of responding to IFN-g if islet architecture were removed. There is plenty of indication from the literature that beta cell heterogeneity exists, and this study does not raise any doubt in my opinion.  Finally, the authors refer to PD-L1 on line 52 as a checkpoint inhibitor—it is not.  It is a checkpoint protein.  Perhaps the most important finding in this study, in my opinion, is the observation that all cell types of the islet exhibit a similar transcriptional response to IFN-g.

Author Response

Reviewer 3

In the study by De Burghgrave et al., the authors performed incubations of islets in vitro with IFN-g and show that all cell types within the islet respond with similar gene expression patterns and that among beta cells, those that are situated in the periphery are the ones most likely to respond to IFN-g.  These data are consistent with the notion that diffusability of compounds and proteins does not occur freely or uniformly within the islet.  The authors correlate these findings in vitro to findings from pancreas sections from NOD mice, wherein expression of PD-L1 is observed in regions adjacent to insulitis.  Overall, the study appears well performed and designed, and the interpretation of the data is appropriate.  

We thank the reviewer for writing that our study is “well performed and designed, and the interpretation of the data is appropriate”.

However, messaging in this study is not unexpected or particularly new, unless the authors were to show that the properties of diffusion are different or unique for IFN-g (compared to another cytokine, e.g. IL-1b). 

We now included in the discussion section on page 14 that it will be interesting to compare the diffusion of IFN-g to the one of other cytokines such as for example IL-1b). We added the following sentence ‘’ It will also be interesting to test whether other cytokines (Il-1β, TNFα) show similar gradient-like action ‘’.

The comment at the end of the results section does not seem to be supported by the data in this study (“the notion of beta cell heterogeneity should be taken with care”)—it is not clear why the authors are raising a doubt regarding heterogeneity, simply because all beta cells are capable of responding to IFN-g if islet architecture were removed. There is plenty of indication from the literature that beta cell heterogeneity exists, and this study does not raise any doubt in my opinion.

In our message, we did not attempt to raise doubts on beta cell heterogeneity. In facts, we pushed the idea that differential beta cell sensitivity to IFN-g is not linked to beta cell subpopulations. We understand from the reviewer that our message was unclear and we decided to remove the sentence “the notion of beta cell heterogeneity should be taken with care” at the end of the result section, and ‘’ especially when it comes to study endocrine cells heterogeneity ‘’ from the end of the discussion.

  Finally, the authors refer to PD-L1 on line 52 as a checkpoint inhibitor—it is not.  It is a checkpoint protein.

We do agree and changed checkpoint inhibitor for checkpoint protein on page 2.

Perhaps the most important finding in this study, in my opinion, is the observation that all cell types of the islet exhibit a similar transcriptional response to IFN-g.

Round 2

Reviewer 2 Report

The authors have answered in an adequate way my main concerns